# Overexpression of Nitrate Transporter *OsNRT2.1* Enhances Nitrate-Dependent Root Elongation

**DOI:** 10.3390/genes10040290

**Published:** 2019-04-09

**Authors:** Misbah Naz, Bingbing Luo, Xueya Guo, Bin Li, Jingguang Chen, Xiaorong Fan

**Affiliations:** 1State Key Laboratory of Crop Genetics and Germplasm Enhancement, MOA Key Laboratory of Plant Nutrition and Fertilization in Low-Middle Reaches of the Yangtze River, Nanjing Agricultural University, Nanjing 210095, China; raymisbah@ymail.com (M.N.); 2016203044@njau.edu.cn (B.L.); 2017103097@njau.edu.cn (X.G.); xinyangzhanlibin@outlook.com (B.L.); 2CAAS-IRRI Joint Laboratory for Genomics-Assisted Germplasm Enhancement, Agricultural Genomics Institute in Shenzhen, Chinese Academy of Agricultural Sciences, Shenzhen 518000, China

**Keywords:** *OsNRT2.1*, NO_3_^−^, auxin transport, root length, rice

## Abstract

Root morphology is essential for plant survival. NO_3_^−^ is not only a nutrient, but also a signal substance affecting root growth in plants. However, the mechanism of NO_3_^−^-mediated root growth in rice remains unclear. In this study, we investigated the effect of *OsNRT2.1* on root elongation and nitrate signaling-mediated auxin transport using *OsNRT2.1* overexpression lines. We observed that the overexpression of *OsNRT2.1* increased the total root length in rice, including the seminal root length, total adventitious root length, and total lateral root length in seminal roots and adventitious roots under 0.5-mM NO_3_^−^ conditions, but not under 0.5-mM NH_4_^+^ conditions. Compared with wild type (WT), the ^15^NO_3_^−^ influx rate of *OsNRT2.1* transgenic lines increased by 24.3%, and the expressions of auxin transporter genes (*OsPIN1a/b/c* and *OsPIN2*) also increased significantly under 0.5-mM NO_3_^−^ conditions. There were no significant differences in root length, ß-glucuronidase (GUS) activity, and the expressions of *OsPIN1a/b/c* and *OsPIN2* in the *pDR5::GUS* transgenic line between 0.5-mM NO_3_^−^ and 0.5-mM NH_4_^+^ treatments together with N-1-naphthylphalamic acid (NPA) treatment. When exogenous NPA was added to 0.5-mM NO_3_^−^ nutrient solution, there were no significant differences in the total root length and expressions of *OsPIN1a/b/c* and *OsPIN2* between transgenic plants and WT, although the ^15^NO_3_^−^ influx rate of *OsNRT2.1* transgenic lines increased by 25.2%. These results indicated that *OsNRT2.1* is involved in the pathway of nitrate-dependent root elongation by regulating auxin transport to roots; i.e., overexpressing *OsNRT2.1* promotes an effect on root growth upon NO_3_^−^ treatment that requires active polar auxin transport.

## 1. Introduction

Nitrogen (N) is an essential macronutrient for plant growth and crop productivity [1]. Plant roots can absorb various forms of nitrogen, including nitrate (NO_3_^−^), ammonium (NH_4_^+^), and organic molecules, which are mainly amino acids [2]. Traditionally, rice is cultivated under flooding conditions. NH_4_^+^ is the primary form in the paddy fields, and rice prefers NH_4_^+^ to NO_3_^−^ [3]. NH_4_^+^ is absorbed into plants by ammonium transporters (AMTs) [1,2]. Excessive NH_4_^+^ in soil is considered to be toxic to rice [1]. However, the concentration of NH_4_^+^ is generally lower than that of NO_3_^−^, and NH_4_^+^ is converted to nitrite and NO_3_^−^ in well-drained soil. NO_3_^−^ is usually the most abundant nitrogen source in aerobic soil; however, this anionic form is soluble in soil water and easy to migrate in soil [4,5]. NO_3_^−^ is very important for rice to improve nitrogen use efficiency and grain yield. NO_3_^−^ is not only a nutrient, but also a signal substance affecting plant growth and development in plants, including inducing the expressions of auxin-related genes [6,7,8,9,10], breaking seed dormancy [11,12], regulating leaf growth [13,14], modulating flowering time [15], and regulating root architecture [16,17,18,19,20,21]. Auxin plays a central role in modulating every step in root growth [22,23,24,25,26,27]. Many studies have shown that the regulation of root system architecture by NO_3_^−^ involves the interaction between NO_3_^−^ and the auxin signaling pathway [16,28,29,30,31].

NO_3_^−^ is absorbed by roots through NO_3_^−^ transporters and then transported to the whole plant, or it combines with carbon to produce amino acids before redistribution [1,32]. The concentration of NO_3_^−^ in soil fluctuates greatly in time and space, and it can reach a range of 1 to 10 mM from few µM after fertilization or nitrification [32,33]. The NO_3_^−^ uptake system in higher plants is composed of the low-affinity transport system (LATS) and the high-affinity transport system (HATS) [1,2,34,35,36]. Rice grows in submerged soils with a low concentration of nitrate; NO_3_^−^ uptake by roots is dominated by high-affinity transport in rice [1,2]. As high-affinity nitrate transporters, NRT2s include five members in rice [37]. Among them, *OsNRT2.1*, *OsNRT2.2*, and *OsNRT2.3a* need a partner protein *OsNAR2.1* to absorb and transport NO_3_^−^ in rice [37,38].

*NRT1.1* participates in NO_3_^−^ uptake by roots under high and low concentrations of NO_3_^−^ [39,40]. *NRT1.1* plays a major role in regulating NO_3_^−^ in root system architecture, because it modulates root growth in response to NO_3_^−^ [10,41,42,43]. Nitrate modulates root development by regulating NRT1.1 [10,19,42,43,44,45]. *NRT2.1* is a high-affinity nitrate transporter [42,46,47], and plays an important role in regulating root development under low concentrations of NO_3_^−^ in *Arabidopsis thaliana* [17,18]. In the previous study, we have reported that the knockdown of *OsNAR2.1* inhibits lateral root formation by reducing auxin transport from shoots to roots under low concentrations of NO_3_^−^, and *OsNAR2.1* might be involved in both NO_3_^−^ uptake and NO_3_^−^ signaling [30].

Auxin transport is mediated by auxin influx carriers (AUX1/LAX family) and efflux carriers (PINs and ABCB/PGPs) in plants [48,49,50,51]. Moreover, PIN proteins regulate auxin gradients during the growth of lateral roots (LRs) [52]. *PIN1* is localized in the basal (root apex-facing) side of the root vasculature [53,54], *PIN2* is expressed in the basal side of the cortical cells and in the apical (shoot apex-facing) side of the epidermal and root cap cells [55,56], *PIN3* is distributed in the columella cells in roots in an apolar manner, *PIN4* is localized in the basal side of cells in the central root meristem with less pronounced polarity in the quiescent center, and *PIN7* is localized in the basal side of the stele cells and in columella cells in an apolar manner [52,57,58].

Our previous results showed that the overexpression of *OsNRT2.1* significantly improves NO_3_^−^ uptake and rice growth under conditions with a low concentration of NO_3_^−^ [47,59]. In this study, we investigated the effect of *OsNRT2.1* on root elongation through nitrate signaling-mediated auxin transport by testing root development, the effects of N-1-naphthylphalamic acid (NPA) on nitrate uptake, and the expression of *OsPINs* in *OsNRT2.1* overexpression lines.

## 2. Materials and Methods

### 2.1. Plant Materials and Growth Conditions

*OsNRT2*.1 transgenic lines (OE1, OE2, and OE3) and the *pDR5::GUS* transgenic line have been described in Luo et al. [59] and Chen et al. [60], respectively. We got the F1 generation seeds of the crossing line of *OE3/DR5::GUS OsNRT2.1* transgenic lines.

Plants were grown in the greenhouse under natural light at day/night temperatures of 30 °C/22 °C and 60% relative humidity. The seeds of wild type (WT) and transgenic lines were surface-sterilized with 10% (*v*/*v*) hydrogen peroxide solution for 30 min, thoroughly rinsed, and washed six times with deionized water. Seeds of uniform size were germinated on the top cover of a 1-L pot with 40 holes per top cover and one seed per hole. The containers were filled with a quarter concentration of nutrient solution with 0.25 mM of NH_4_NO_3_, 0.5 mM of NH_4_^+^ or 0.5 mM of NO_3_^−^. Ten-day-old seedlings grown in 0.5 mM of NH_4_^+^ or 0.5 mM of NO_3_^−^ nutrient solution were transplanted into holes on the top cover of a 7-L pot containing the corresponding nutrient solution, respectively.

The complete nutrient solution of the International Rice Research Institute (IRRI) included 1.0 mM of MgSO_4_·7H_2_O, 1.0 mM of CaCl_2_, 0.5 mM of Na_2_SiO_3_, 0.35 mM of K_2_SO_4_, 0.3 mM of KH_2_PO_4_, 20.0 μM of Fe-EDTA (ethylene diaminetetra acetic acid tetrasodium salt), 20.0 μM of H_3_BO_3_, 9.0 μM of MnCl_2_, 0.77 μM of ZnSO_4_, 0.39 μM of (NH_4_)_6_Mo_7_O_24_, and 0.32 of μM CuSO_4_, with pH 5.5. The nutrient solution was replaced once every day. NH_4_^+^ and NO_3_^−^ supplied in the nutrient solution were (NH_4_)_2_SO_4_ and Ca(NO_3_)_2_, respectively. Ca^2+^ in NH_4_^+^ nutrient solution was supplemented with CaCl_2_, and the same concentration of Ca^2+^ was added under different treatments. Nitrification inhibitor dicyandiamide was added to each pot to prevent the oxidation of nutrient solution.

### 2.2. qRT-PCR Analysis

Total RNA was extracted using TRIzol reagent (Vazyme Biotech Co, Ltd., Nanjing, China). DNase I-treated total RNAs were used for reverse transcription (RT) with HiScript II Q Select RT SuperMix for qPCR (+gDNA wiper) kit (Vazyme Biotech Co). Quantitative assays were performed in triplicate using the 2 × T5 Fast qPCR Mix (SYBRGreenI) kit (Vazyme Biotech Co). The primers for qRT-PCR are shown in Appendix A.

### 2.3. Assay of the Concentration of Total Nitrogen

About 0.05 g of crushed dry samples were digested with H_2_SO_4_-H_2_O_2_ at 280 ℃. After cooling, the digested samples were diluted to 100 mL in distilled water. The concentration of total N was measured using the Kjeldahl method [61].

### 2.4. N-1-Naphthylphalamic Acid Treatment

Three-day-old normal grown seedlings were transferred to nutrient solutions containing 1 μM of auxin efflux inhibitor N-1-naphthylphalamic acid (NPA) from 100 mM of NPA that were dissolved in dimethyl sulfoxide(DMSO)and control containing the same amount of DMSO. Sampling was performed after seven days of treatment.

### 2.5. Determination of ^15^NO_3_^−^ Influx Rate in Roots

The influx rates of ^15^NO_3_^−^ in roots were assayed as described previously [62]. Plant seedlings were transferred to 0.5 mM of ^15^NO_3_^−^ (atom% ^15^N: ^15^NO_3_^−^, 99%) nutrient solution with or without 1 μM of NPA for 5 min, and then, the seedlings were transferred to 0.1 mM of CaSO_4_ solution for 1 min before sampling. The concentration of ^15^N was analyzed by isotope ratio mass spectrometry (DELTA V Advantage Isotope Ratio Mass Spectrometer, Thermo Fisher Scientific, Waltham, MA, USA).

### 2.6. Analysis of ß-Glucuronidase Activity

For ß-Glucuronidase (GUS) staining, crown roots were immersed in GUS staining solution (1 mg/mL of X-glucuronide in 100 mM of sodium phosphate, 0.5 mM of ferrocyanide, 0.5 mM of ferricyanide, and 0.1% Triton X-100, pH 7.2), and then incubated at 37 °C in the dark.

For GUS activity assay, 5 μL of extract was added to 450 μL of GUS extraction buffer containing 1 mM of 4-methylumbelliferyl β-d-glucuronide, and incubated at 37 °C. Thereafter, 20 μL of reaction mixture was added into a 180-μL stop solution (1 M of sodium carbonate) for 10 min. Then, the fluorescence values were measured at 365 nm by a Fluorolite 1000 fluorometer (DYNEX technologies, USA).

### 2.7. Root Scanning

The root parameters were measured as previously described [30]. The length of seminal and adventitious roots was measured with a ruler, and the LR density was calculated as the ratio of the root length to the number of roots. The total root length and LR length were measured using the WinRhizo scanner-based image analysis system (Regent Instruments, Montreal, QC, Canada). The total root length represents the sum of the seminal roots, adventitious roots, and all the lateral roots. The total adventitious roots represent the sum of all the adventitious root lengths in every plant. The total LR length in seminal roots is the sum of the lateral root length of the seminal roots in every plant and the total LR length in adventitious roots is the sum of the lateral root length in the adventitious roots of every plant.

### 2.8. Statistical Analysis

The data of the experiments were analyzed by one-way ANOVA and Tukey’s test at *p* < 0.05 to determine the statistically significant differences among different treatments. All the statistical evaluations were performed using SPSS version 20.0 statistical software (SPSS Inc., Chicago, IL, USA).

## 3. Results

### 3.1. Effect of OsNRT2.1 Overexpression on Root Growth under 0.25-mM NH_4_NO_3_ Conditions

In previous studies, three genetically stable transgenic rice lines were obtained by transgenic technology [59]. There were no significant differences in grain length, grain width, and 1000-grain weight between *OsNRT2.1* transgenic lines and WT (Appendix A). The overexpression of *OsNRT2.1* had no significant effect on grain size in rice. Moreover, the concentration of total N in the seeds of *OsNRT2.1* transgenic lines was analyzed, and no significant difference was found between them and WT (Appendix A).

We also performed germination experiments and examined the growth of seedlings under 0.25-mM NH_4_NO_3_ conditions (Figure 1A). There were no significant differences in the final seed germination rate (Figure 1B) and shoot length between transgenic lines and WT on the ninth day (Figure 1C). Since the seventh day, the root length of transgenic lines was significantly longer than that of WT (Figure 1D).

The roots of transgenic lines on the ninth day were scanned and analyzed (Figure 2). Compared with WT, the total root length of transgenic plants under the 0.25-mM NH_4_NO_3_ conditions increased by 42.2% (Figure 3B), including a 45.3% increase in the seminal root length (Figure 3C), a 37.7% increase in the total adventitious root length (Figure 3D), a 48.7% increase in the total lateral root length in seminal roots (Figure 3G), and a 41.7% increase in the total lateral root length in adventitious roots (Figure 3H). However, there were no significant differences in the lateral root density in seminal roots and the lateral root density in adventitious roots between the transgenic lines and WT, respectively (Figure 3E,F).

In addition, we examined the growth of *OsNRT2.1* transgenic seedlings under 0.5-mM NH_4_^+^ (Appendix A) or NO_3_^−^ (Appendix A) conditions. Under 0.5-mM NH_4_^+^ conditions, there were no significant differences in shoot length and root length between the transgenic lines and WT (Appendix A). Under 0.5-mM NO_3_^−^ conditions, the shoot length of transgenic lines was significantly longer than that of WT from the 25th day (Appendix A), and the root length of transgenic lines was significantly longer than that of WT from the 10th day (Appendix A).

The roots of transgenic plants were scanned and analyzed under 0.5-mM NH_4_^+^ (Figure 2B and Figure 3A) or NO_3_^−^ (Figure 2C and Figure 3) conditions on the 10th day. Under 0.5-mM NH_4_^+^ conditions, there was no significant difference in the total root length between the transgenic lines and WT (Figure 3A), including seminal root length (Figure 3B), total adventitious root length (Figure 3C), lateral root density in seminal roots (Figure 3D), lateral root density in adventitious roots (Figure 3E), total lateral root length in seminal roots (Figure 3F), and total lateral root length in adventitious roots (Figure 3G).

Under 0.5-mM NO_3_^−^ conditions, the total root length of transgenic plants increased by 38.6% compared with that of WT (Figure 3A), including a 24.0% increase in seminal root length (Figure 3B), a 36.4% increase in total adventitious root length (Figure 3C), a 46.0% increase in total lateral root length in seminal roots (Figure 3F), and a 39.7% increase in total lateral root length in adventitious roots (Figure 3G). However, there were no significant differences in lateral root density in seminal roots and lateral root density in adventitious roots between the transgenic lines and WT, respectively (Figure 3D,E).

### 3.2. Effect of OsNRT2.1 Overexpression on the Expressions of OsPINs under 0.5-mM NH_4_^+^ or NO_3_^−^ Conditions

The expression of auxin transporter *OsPINs* in roots were also assayed in this study. There were no significant differences in the expression of *OsPIN1a*, *OsPIN1b*, *OsPIN1c*, and *OsPIN2* between the transgenic plants and WT under 0.5-mM NH_4_^+^ conditions (Figure 4A–D). Meanwhile, the expression of *OsPIN1a*, *OsPIN1b*, *OsPIN1c*, and *OsPIN2* in transgenic lines increased significantly compared with those in WT under 0.5-mM NO_3_^−^ conditions (Figure 4A–D).

### 3.3. Effect of NPA on Root Growth under 0.5-mM NH_4_^+^ and NO_3_^−^ Conditions

In order to investigate the effect of auxin transport on nitrate-mediated root growth, the rice seedlings of the *pDR5::GUS* transgenic line were treated with NPA, which is an auxin transport inhibitor. Exogenous NPA treatment significantly reduced the seminal root length under 0.5-mM NH_4_^+^ or NO_3_^−^ conditions, respectively (Figure 5A,C). The seminal root length under 0.5-mM NO_3_^−^ conditions was significantly longer than that under 0.5-mM NH_4_^+^ conditions without NPA treatment (Figure 6A,C). There was no significant difference in rice root length between 0.5-mM NO_3_^−^ and 0.5-mM NH_4_^+^ treatments with NPA treatment (Figure 5A,C).

ß-Glucuronidase staining showed that the color of seminal roots of *pDR5::GUS* transgenic seedlings without NPA treatment was significantly darker under 0.5-mM NO_3_^−^ treatment than under 0.5-mM NH_4_^+^ treatment (Figure 5B). The color of *pDR5::GUS* transgenic seedlings with NPA treatment was very light under 0.5-mM NH_4_^+^ or NO_3_^−^ treatment (Figure 5B). The GUS activity of *pDR5::GUS* transgenic seedlings was further analyzed to confirm the result of GUS staining. NPA could significantly inhibit the GUS activity under 0.5-mM NH_4_^+^ or NO_3_^−^ conditions (Figure 5D). The GUS activity under 0.5-mM NO_3_^−^ was significantly higher than that under 0.5-mM NH_4_^+^ without NPA treatment, and there was no significant difference in the GUS activity between 0.5-mM NH_4_^+^ and 0.5 mM-NO_3_^−^ with NPA treatment (Figure 5D). The same results were found for *OE3/DR5::GUS* transgenic seedlings: the seminal root tip color of the *OE3/DR5::GUS* transgenic line without NPA treatment was significantly stronger under 0.5-mM NO_3_^−^ treatment than under 0.5-mM NH_4_^+^ treatment (Appendix A). The color of *pDR5::GUS* transgenic seedlings with NPA treatment was very light under 0.5-mM NH_4_^+^ or NO_3_^−^ treatment (Appendix A).

The expressions of *OsPIN1a*, *OsPIN1c*, and *OsPIN2* under 0.5-mM NO_3_^−^ was significantly higher than those under 0.5-mM NH_4_^+^ without NPA treatment (Figure 5E–H). The expressions of *OsPIN1a*, *OsPIN1b*, *OsPIN1c*, and *OsPIN2* were significantly inhibited under 0.5-mM NH_4_^+^ or NO_3_^−^ conditions with NPA treatment; however, there were no significant differences in the expressions of *OsPIN1a*, *OsPIN1b*, *OsPIN1c*, and *OsPIN2* under 0.5-mM NH_4_^+^ conditions and under 0.5-mM NO_3_^−^ with NPA treatment (Figure 5E–H).

### 3.4. NPA Inhibits the Effect of OsNRT2.1 on Root Growth under 0.5-mM NO_3_^−^ Conditions

Moreover, NPA was added to *OsNRT2.1* transgenic lines under 0.5-mM NO_3_^−^ conditions (Figure 6A). There were no significant differences in the total root length between transgenic plants and WT (Figure 7B), including the seminal root length (Figure 6C), total adventitious root length (Figure 6D), lateral root density in seminal roots (Figure 6E), lateral root density in adventitious roots (Figure 6F), total lateral root length in seminal roots (Figure 6G), and total lateral root length in adventitious roots (Figure 6H).

Similarly, there were no significant differences in the expressions of *OsPIN1a*, *OsPIN1b*, *OsPIN1c*, and *OsPIN2* between *OsNRT2.1* transgenic lines and WT under 0.5-mM NO_3_^−^ conditions with NPA treatment (Appendix A).

### 3.5. NPA Inhibits the Effect of OsNRT2.1 on ^15^NO_3_^−^ Influx Rate in Roots under 0.5-mM NO_3_^−^ Conditions

The short-term NO_3_^−^ uptake in roots of the transgenic lines and WT was analyzed by exposing the seedlings to 0.5 mM of ^15^NO_3_^−^ for 5 min, so as to determine the effect of *OsNRT2.1* overexpression on root NO_3_^−^ influx to the whole plants. Compared with WT, the ^15^NO_3_^−^ influx rate of *OsNRT2.1* transgenic lines increased by 24.3% (Appendix A [59] and Appendix A).

The ^15^NO_3_^−^ influx rate of the transgenic lines under 0.5-mM ^15^NO_3_^−^ conditions with NPA treatment was also analyzed. The results showed that WT and all the rice overexpression lines decreased the ^15^NO_3_^−^ influx rate with NPA treatment [Appendix A]. These data suggested that the *NRT2.1*-related import of NO_3_^−^ may be partially auxin-dependent. However, compared with WT, the ^15^NO_3_^−^ influx rate of *OsNRT2.1* transgenic lines under 0.5-mM ^15^NO_3_^−^ conditions with NPA treatment was higher by 25.2% (Figure 7 and Appendix A).

## 4. Discussion

In plants, NO_3_^−^ is not only a nutrient, but also a signal substance affecting root growth [16,17,18,19,20,21,30,31,63,64]. However, the mechanism of NO_3_^−^-mediated root growth in rice remains unclear.

We observed that the overexpression of *OsNRT2.1* increased the total root length in rice, including the seminal root length, total adventitious root length, and total lateral root length in seminal roots and adventitious roots under 0.25-mM NH_4_NO_3_ conditions (Figure 1 and Figure 2). *OsNRT2.1* overexpression lines have longer seminal roots under NO_3_^−^, while their respective LR number was not increased (Figure 3). That would indicate there are more LRs in seminal roots following NO_3_^−^ treatment, which was in agreement with other plant species [16,65]. Further analysis showed that the total root length of transgenic plants increased by 38.6% compared with WT under 0.5-mM NO_3_^−^ conditions (Figure 3A), including a 24.0% increase in seminal root length (Figure 3B), a 36.4% increase in total adventitious root length (Figure 3C), a 46.0% increase in total lateral root length in the seminal roots (Figure 3F), and a 39.7% increase in total lateral root length in the adventitious roots (Figure 3G). However, there was no significant difference between the transgenic lines and WT under 0.5-mM NH_4_^+^ conditions (Appendix A, Figure 2 and Appendix A). The regulation of NO_3_^−^ on plant root growth is not exerted by the direct perception of external NO_3_^−^, but rather depends on the amount of NO_3_^−^ absorbed by plants [16,30,65]. The short-term NO_3_^−^ uptake in roots of the transgenic lines and WT was analyzed by exposing the plants to 0.5 mM of ^15^NO_3_^−^ for 5 min, in order to determine the effect of *OsNRT2.1* overexpression on root NO_3_^−^ influx into the whole plants. Compared with WT, the ^15^NO_3_^−^ influx rate of *OsNRT2.1* transgenic lines increased by 24.3% (Appendix A [59] and Appendix A). Overall, these results suggested that the overexpression of *OsNRT2.1* might promote root elongation by increasing NO_3_^−^ uptake in rice.

Many studies have shown that the regulation of root system architecture by NO_3_^−^ involves a strong interaction between NO_3_^−^ and the auxin-signaling pathway [16,28,29,30,31]. Auxin plays a central role in modulating every step of root growth [22,23,24,25,26,27]. There were no significant differences in the expressions of *OsPIN1a*, *OsPIN1b*, *OsPIN1c*, and *OsPIN2* between transgenic plants and WT under 0.5-mM NH_4_^+^ conditions (Figure 4A–D). However, the expressions of *OsPIN1a, OsPIN1b*, *OsPIN1c*, and *OsPIN2* in transgenic lines increased significantly compared with WT under 0.5-mM NO_3_^−^ conditions (Figure 4A–D). An analysis of gene structure showed that the distribution of *OsPIN1a*, *OsPIN1b*, and *OsPIN1c* in rice is similar to that of *AtPIN1* [66]. AtPIN1 is highly conserved and plays an important role in auxin polar transport [48]. AtPIN1 mediates the vertical transport of IAA from shoot to root along the embryonic apical-basal axis [49,50]. *OsPIN1a* and *OsPIN1c* are expressed in the lateral root cap region, *OsPIN1b* is expressed in the root cap, and *OsPIN1b* and *OsPIN1c* are expressed in the meristem [66]. OsPIN1a is an auxin efflux protein that regulates the negative phototropism in rice roots [53]. *OsPIN1b* regulates root growth and seminal root elongation; the adventitious roots of *ospin1b* RNAi transgenic plants are significantly reduced compared with WT, and the seminal roots of *ospin1b* T-DNA mutants are shorter than those of WT plants [54,67]. *OsPIN2* is expressed in the lateral root cap region, root epidermal, and outer cortex cells [55,66,68]. The overexpression of *OsPIN2* results in reduced auxin levels in the rice root tip [60]. Wang et al. [55] recently reported that *OsPIN2* plays an important role in mediating root gravitropic responses and is essential for plants to produce normal root growth angle in rice. *OsPIN2* is an auxin efflux carrier protein, and may regulate the flow of auxin from the root tip to the root elongation zone and the distribution of auxin in seminal root tips, thereby modulating the seminal root elongation and lateral root formation in rice [56]. Therefore, *OsNRT2.1* might be involved in the auxin transport pathway, regulating root growth and development under NO_3_^−^ supplied conditions.

In order to investigate the effect of auxin transport on nitrate-mediated root growth, the *pDR5::GUS* transgenic seedlings were treated with NPA, which is an auxin transport inhibitor. The *pDR5::GUS* reporter system is a sensitive and simple system for monitoring auxin response and distribution in plant cells [57,60,69]. The seminal root length of *pDR5::GUS* transgenic seedlings under 0.5-mM NO_3_^−^ conditions was significantly longer than that under 0.5-mM NH_4_^+^ conditions without NPA treatment (Figure 5A,C Appendix A). Sun et al. [51] showed that the auxin levels in LRs and RT were higher under NO_3_^−^ conditions than those under NH_4_^+^ conditions, indicating that auxin distribution in roots may be regulated by NO_3_^−^ supply, which can increase the polar auxin transport to improve the seminal root length. Exogenous NPA treatment could significantly reduce the seminal root length under 0.5-mM NH_4_^+^ or NO_3_^−^ conditions, respectively (Figure 5A,C and Appendix A). There was no significant difference in root length under 0.5-mM NO_3_^−^ conditions and under 0.5-mM NH_4_^+^ conditions with NPA treatment (Figure 5A,C and Appendix A). Without NPA treatment, the expressions of *OsPIN1a, OsPIN1c*, and *OsPIN2* under 0.5-mM NO_3_^−^ conditions was significantly higher than those under 0.5-mM NH_4_^+^ conditions (Figure 5E–H). In addition, the expressions of *OsPIN5a*, *OsPIN5b*, *OsPIN5c*, *OsPIN8*, *OsPIN9*, *OsPIN10a*, and *OsPIN10b* were also increased under 0.5-mM NO_3_^−^ conditions [51]. The expressions of *OsPIN1a*, *OsPIN1b*, *OsPIN1c*, and *OsPIN2* was significantly inhibited under 0.5-mM NH_4_^+^ or NO_3_^−^ conditions with NPA treatment (Figure 5E–H). Exogenous NPA treatment could significantly inhibit the GUS activity under 0.5-mM NH_4_^+^ or NO_3_^−^ conditions (Figure 5D). The GUS activity under 0.5-mM NO_3_^−^ conditions was significantly higher than that under 0.5-mM NH_4_^+^ conditions without NPA treatment, and there was no significant difference in GUS activity under 0.5-mM NH_4_^+^ conditions and under 0.5-mM NO_3_^−^ conditions with NPA treatment (Figure 5B,D and Appendix A). However, the DR5 signal was increased in the root tip of the “Nanguang” variety of seedlings under NH_4_^+^ conditions with the addition of NPA compared with that without NPA [70]. This could be attributed to the difference between rice varieties. These results indicated that nitrate could be used as a signal substance to induce the expressions of *OsPIN1a/b/c* and *OsPIN2*, regulate auxin transport to roots, and promote root elongation. However, nitrate could not induce root growth when auxin transport was inhibited.

Plants adjust their growth and development in response to changing environmental conditions by sensing external signals and integrating them into plant hormone-signaling pathways such as the auxin-signaling pathway [29,71,72,73,74,75,76]. The root growth of the auxin-insensitive mutant *axr4* cannot be promoted by adding NO_3_^−^ [16]. Different environmental signals and endogenous signals can regulate auxin distribution by affecting polar transport [66,77]. When NPA was added to 0.5 mM of NO_3_^−^ nutrient solution, there was no significant difference in the total root length between transgenic plants and WT (Figure 6). Similarly, there was no significant difference in the expressions of *OsPIN1a/b/c* and *OsPIN2* between *OsNRT2.1* transgenic plants and WT under 0.5-mM NO_3_^−^ conditions with NPA treatment (Appendix A), although the ^15^NO_3_^−^ influx rate of *OsNRT2.1* transgenic lines under 0.5-mM ^15^NO_3_^−^ conditions with NPA treatment increased by 25.2% (Figure 7 and Appendix A). These results indicated that *OsNRT2.1* promoted root elongation by regulating auxin transport to roots, when auxin transport was inhibited; only increasing NO_3_^−^ uptake could not promote root growth in rice.

However, the overexpression of *OsNRT2.1* increased NO_3_^−^ influx rate, induced the expressions of *OsPINs* in turn with the increase of the nitrate influx, and up-regulated auxin transport to roots, resulting in enhanced root development. Based on the above results in overexpression lines, the seedlings showed greater growth and yield than WT under low concentrations of nitrate in hydroponic solutions (Appendix A) and low concentrations of nitrogen fertilizer in the field [59]. Thus, this study provides clear evidence for the super phenotype of overexpression lines from the perspective of the root.

## 5. Conclusions

NO_3_^−^ could be used as a signal substance to induce the expressions of *OsPIN1a/b/c* and *OsPIN2*, regulate auxin transport to the root system, and promote root elongation. We first report that the overexpression of *OsNRT2.1* can show great root growth phenotypes under low NO_3_^−^ conditions through the modulation of auxin transport, and root elongation may depend on auxin transport, but not the increase of nitrate influx.

## Figures and Tables

**Figure 1 genes-10-00290-f001:**
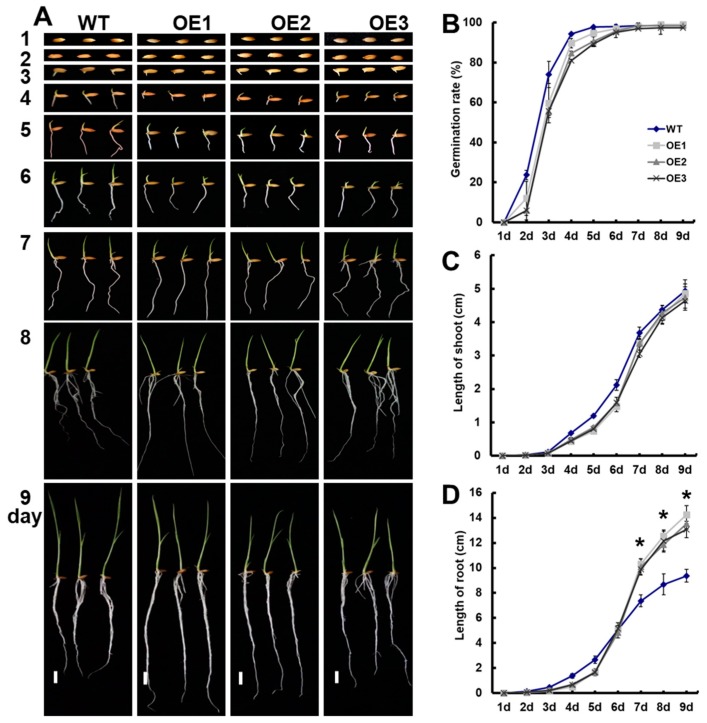
Overexpression of *OsNRT2.1* affects root growth at the seedling stage under 0.25-mM NH_4_NO_3_ conditions. Rice seedlings were grown in a quarter concentration of nutrient solution containing 0.25 mM of NH_4_NO_3_ from the beginning. The nutrient solution was replaced daily. (**A**) Seed germination in wild type (WT) and transgenic plants. Scale bars = 1 cm. (**B**) Seed germination rate, (**C**) shoot length, and (**D**) root length of WT and transgenic plants. Error bars: SE (*n* = 10). Significant differences between transgenic lines and WT are indicated by asterisks (*p* < 0.05, one-way ANOVA).

**Figure 2 genes-10-00290-f002:**
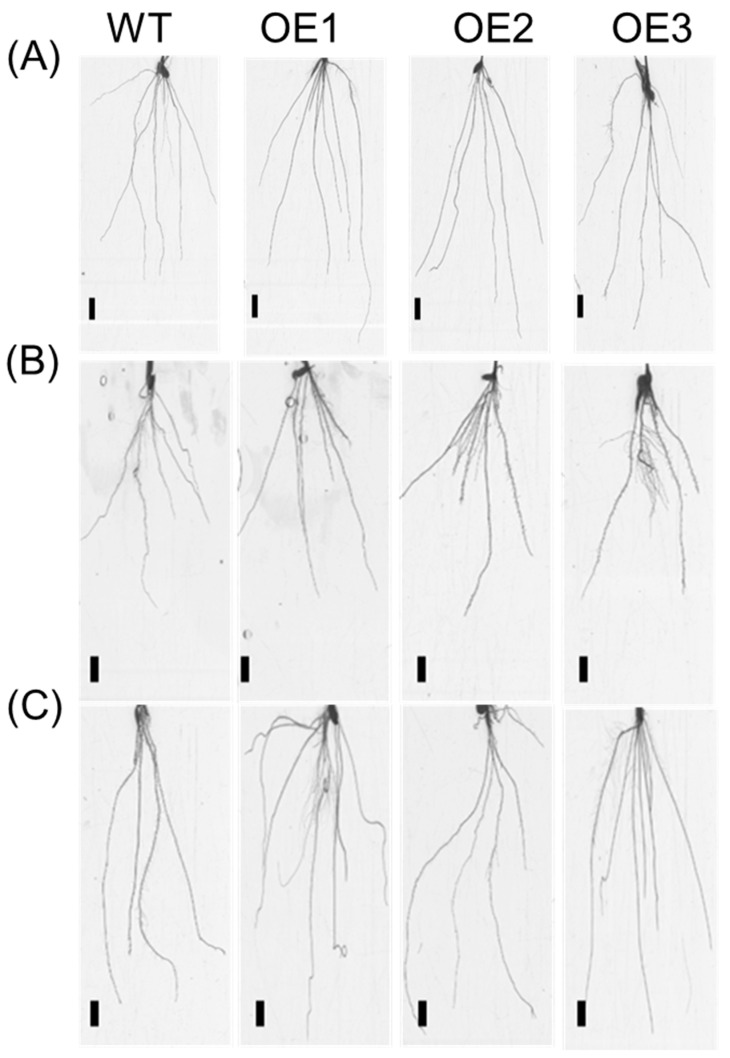
Root morphology of WT and *OsNRT2.1* transgenic lines (**A**) under 0.25-mM NH_4_NO_3_ conditions, (**B**) under 0.5-mM NH_4_^+^ conditions, (**C**) under 0.5-mM NO_3_^−^ conditions. WT and transgenic lines on the ninth day were shown in Figure 1 (bar = 1 cm).

**Figure 3 genes-10-00290-f003:**
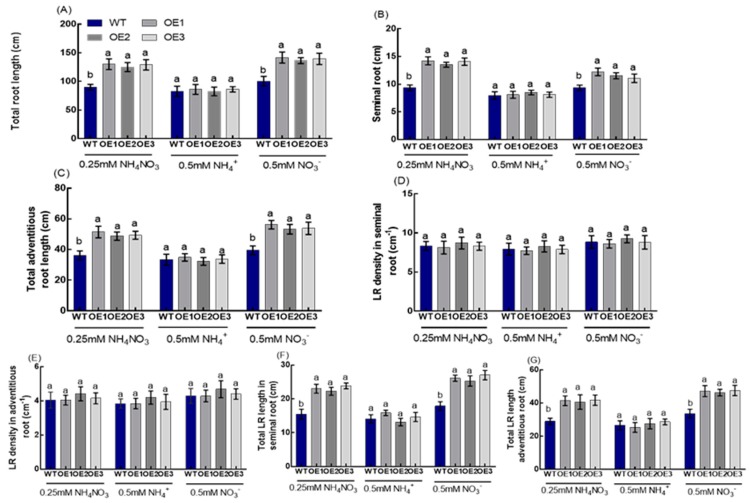
Root morphology of WT and OsNRT2.1 transgenic lines under 0.25-mM NH_4_NO_3_, 0.5-mM NH_4_^+^, or 0.5-mM NO_3_^−^ conditions. WT and transgenic plants on the 10th day were shown in Appendix A. (**A**) Total root length, (**B**) seminal root length, (**C**) total adventitious root length, (**D**) LR density in seminal roots, (**E**) lateral root (LR) density in adventitious roots, (**F**) total LR length in seminal roots, and (**G**) total LR length in adventitious roots. Total root length is the sum of the seminal root, adventitious roots, and all lateral roots. Error bars: SE (*n* = 5). Significant differences between transgenic lines and WT are indicated by different letters (*p* < 0.05, one-way ANOVA).

**Figure 4 genes-10-00290-f004:**
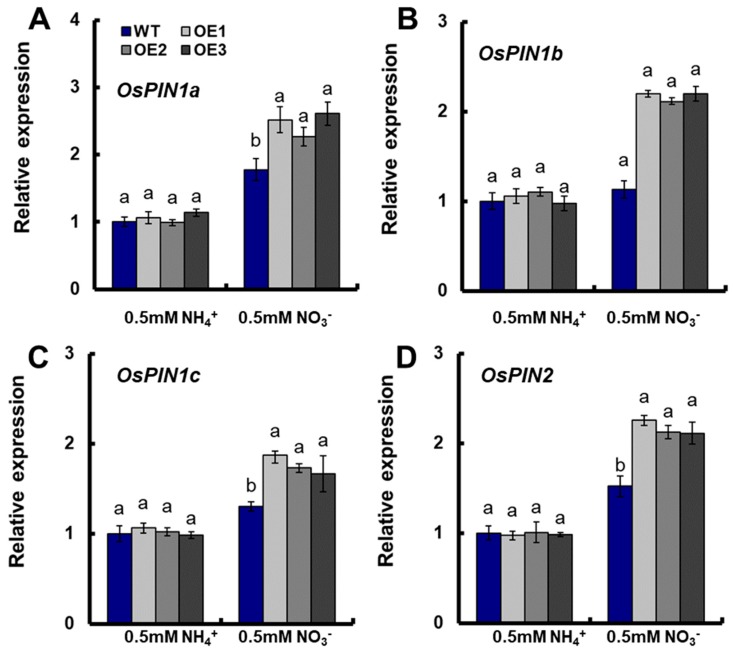
The expressions of *OsPINs* in the roots of WT and *OsNRT2.1* transgenic lines under 0.5-mM NH_4_^+^ or 0.5 mM NO_3_^−^ conditions. WT and transgenic plants on the 10th day were shown in Figure 3. Real-time quantitative RT-PCR analysis of the expressions of (**A**) *OsPIN1a*, (**B**) *OsPIN1b*, (**C**) *OsPIN1c*, and (**D**) *OsPIN2* in WT and *OsNRT2.1* transgenic lines. RNA was extracted from roots. Error bars: SE (*n* = 5). Significant differences between transgenic lines and WT are indicated by different letters (*p* < 0.05, one-way ANOVA).

**Figure 5 genes-10-00290-f005:**
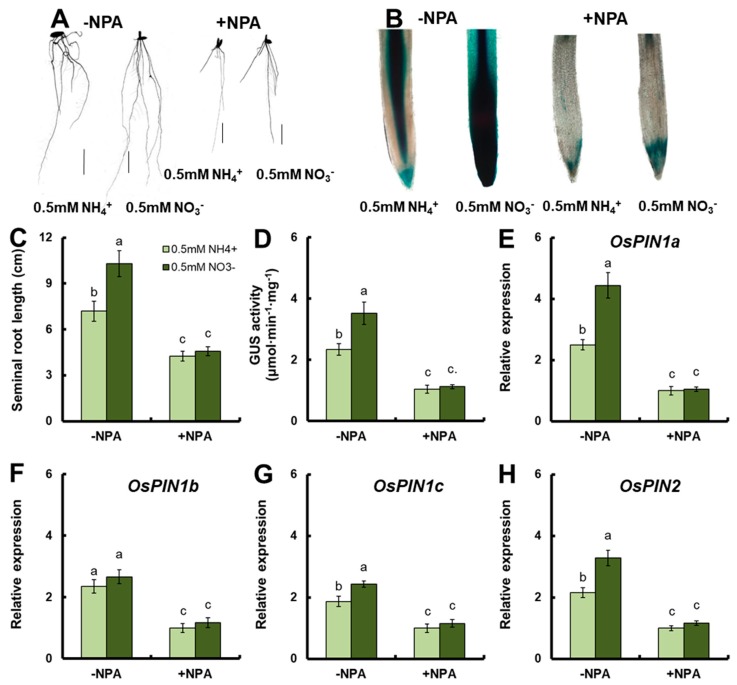
Effects of N-1-naphthylphalamic acid (NPA) on root growth under 0.5-mM NH_4_^+^ or 0.5 mM NO_3_^−^ conditions. Three-day-old normal grown *pDR5::GUS* transgenic seedlings were transferred to nutrient solutions containing 1 μM of NPA from 100 mM of NPA that was dissolved in DMSO (dimethyl sulfoxide), and the control containing the same amount of DMSO (dimethyl sulfoxide). Sampling was performed after seven days of treatment. (**A**) Root morphology of rice plants under 0.5-mM NH_4_^+^ or 0.5-mM NO_3_^−^ conditions with or without NPA treatment. (bar = 1 cm). (**B**) ß-Glucuronidase (GUS) expression in the root tips of *pDR5::GUS* seedlings under 0.5-mM NH_4_^+^ or 0.5 mM NO_3_^−^ conditions with or without NPA treatment. Root tips were stained for 2 h at 37 °C in the dark. (**C**) Seminal root length. (**D**) GUS activity of roots. Real-time quantitative RT-PCR analysis of the expressions of (**E**) *OsPIN1a*, (**F**) *OsPIN1b*, (**G**) *OsPIN1c*, and (**H**) *OsPIN2* in *pDR5::GUS* seedlings under 0.5-mM NH_4_^+^ or 0.5-mM NO_3_^−^ conditions with or without NPA treatment. RNA was extracted from roots. Error bars: SE (*n* = 5). Significant differences between transgenic lines and WT are indicated by different letters (*p* < 0.05, one-way ANOVA).

**Figure 6 genes-10-00290-f006:**
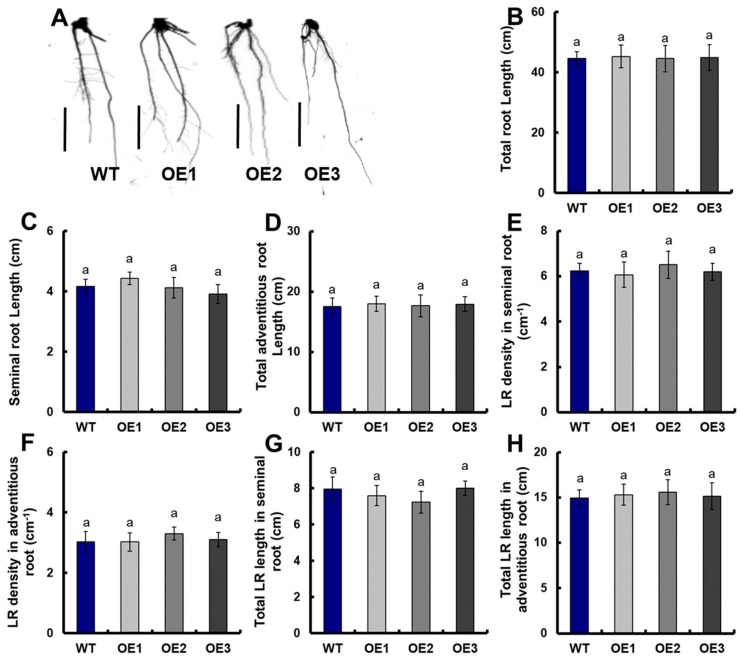
Root morphology of WT and *OsNRT2.1* transgenic lines with NPA treatment under 0.5 mM of NO_3_^−^. Three-day-old normal grown WT and transgenic seedlings were transferred to nutrient solutions containing 1 μM of NPA from 100 mM of NPA that was dissolved in DMSO and the control containing the same amount of DMSO. Sampling was performed after seven days of treatment. (**A**) Root morphology (bar = 1 cm), (**B**) total root length, (**C**) seminal root length, (**D**) total adventitious root length, (**E**) LR density in seminal roots, (**F**) LR density in adventitious roots, (**G**) total LR length in seminal roots, and (**H**) total LR length in adventitious roots. Total root length is the sum of the seminal roots, adventitious roots, and all lateral roots. Error bars: SE (*n* = 5). Significant differences between transgenic lines and WT are indicated by different letters (*p* < 0.05, one-way ANOVA).

**Figure 7 genes-10-00290-f007:**
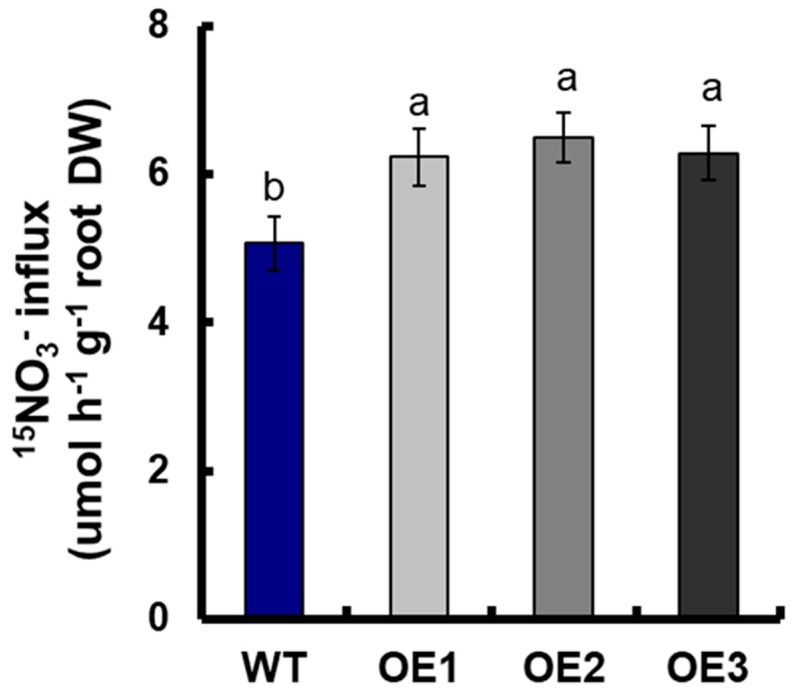
^15^NO_3_^−^ influx rate in roots of WT and *OsNRT2.1* transgenic lines under 0.5-mM ^15^NO_3_^−^ conditions with NPA treatment. WT and transgenic plants under 0.5-mM NO_3_^−^ with 1 μM of NPA treatment are shown. The plants were transferred to a quarter concentration of nutrient solution containing 0.5 mM of ^15^NO_3_^−^ and 1 μM of NPA for 5 min. Error bars: SE (*n* = 5). Significant differences between transgenic lines and WT are indicated by different letters (*p* < 0.05, one-way ANOVA).

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
