# Peer review of "Overexpression of Nitrate Transporter OsNRT2.1 Enhances Nitrate-Dependent Root Elongation"

_genes, 2019, doi:10.3390/genes10040290_

Round 1
Reviewer 1 Report
Overexpression of nitrate transporter OsNRT2.1 enhances nitrate-dependent root elongation
Naz et al
This study is a follow-up to previous work from the same group that showed overexpression of OsNRT2.1 nitrate uptake and growth in low nitrate conditions. This manuscript is novel as they show this affect is linked to auxin transport.
They demonstrate that overexpression of OsNRT2.1 increases root biomass in response to growth in the presence of 0.25mM and 0.5mM NO3-. This effect is removed in the presence of the auxin transport inhibitor NPA. Indeed NPA removes the increased in root growth observed when wildtype plants are growth in the presence of 0.5mM NO3-. These phenotypic changes are mimicked with the expression of OsPIN1a/b/c and OsPIN2.
However NPA had no affect on the increased influx of 15NO3- into transgenic roots.
This is a well-performed study and the data is well presented. The authors could even speculate more about the importance of their findings in the context of food security in rice-producing areas that have low nitrate.
These are a few small comments:
Line 21: Conditions not condition
Line 41: expression of related genes. Related to what?
Line 54, 60: What is OsNAR2.1? This hasn’t been mentioned yet.
Paragraph beginning line 54: This jumps between genes in different sentences. It needs to be rewritten so that it is easier to follow.
Line 125: grain width
Paragraph beginning line 125: This is an unusual way to begin the results section. Can the author explain the creation of the transgenic line either in detail or by including a reference to their creation in a previous study.
Line 130: Performed.
Line 143: Could the authors define ‘total root length’ for the benefit of naïve readers
Line 193: Changed font size.
Figure S2: Has this graph been previously published? It is similar to [47, Figure 3B]. This should be mentioned in the text.
Line 240 (and other places): The expression… Not The expressions
Discussion/ Conclusion: Could the authors comment on the physiological relevance of their findings? Does the activity of OsNRT2.1 allows auxin to stimulate root growth that in turn allows the uptake of more nitrate? Is this an effective response to growth of rice in low nitrate conditions? Could OsNRT2.1 overexpression be used to improve rice yields under low nitrate conditions
Author Response
Point 1: Line 21: Conditions not condition
Response 1: Ok, we have corrected this error. Please see the line 23 in the revised version with trackers.
Point 2: Line 41: expression of related genes. Related to what?
Response 2: We have made it clear that auxin related genes. Please see the line 51 in the revised version with trackers.
Point 3: Line 54, 60: What is OsNAR2.1? This hasn’t been mentioned yet?
Response 3: NRT2s family need partner protein OsNAR2.1 to uptake and transport NO3- in rice. We have made this gene clear. Please see the line 66. in the revised version with trackers.
Point 4: Paragraph beginning line 54: This jumps between genes in different sentences. It needs to be rewritten so that it is easier to follow.
Response 4: We have rewritten it. Please see the lines from 64 to 66. in the revised version with trackers.
Point 5: Line 125: grain width
Response 5: We have corrected this error. Please see the line 160 in the revised version with trackers.
Point 6: Paragraph beginning line 125: This is an unusual way to begin the results section. Can the author explain the creation of the transgenic line either in detail or by including a reference to their creation in a previous study?
Response 6: We have published a paper about the overexpression of OsNRT2.1 rice line in Frontier in Plant Science journal last year. So, we didn’t describe it in detail and just cited the reference. Please see the lines 95,96, 159 and 160 in the revised version with trackers.
Point 7: Line 130: Performed.
Response 7: We have corrected this error. Please see the line 165 in the revised version with trackers.
Point 8: Line 143: Could the authors define ‘total root length’ for the benefit of naïve readers
Response 8: The “total root length” includes the sum of seminal root, adventitious roots and all lateral roots. We have noted it in method and very figure legend. Please see the lines from 149 to 152, from 231 to 238 and from 318 to 320.
Point 9: Line 193: Changed font size.
Response 9: Ok, thank you very much!
Point 10: Figure S2: Has this graph been previously published? It is similar to [47, Figure 3B]. This should be mentioned in the text.
Response 10: Yes, we have published this data last year under -NPA, here we cited this data from reference 47 and mentioned the reference in text. Please see the lines from 327 to 330 in the revised version with trackers.
Point 11: Line 240 (and other places): The expression… Not The expressions
Response 11: We have corrected this error.
Point 12: Discussion/ Conclusion: Could the authors comment on the physiological relevance of their findings? Does the activity of OsNRT2.1 allows auxin to stimulate root growth that in turn allows the uptake of more nitrate? Is this an effective response to growth of rice in low nitrate conditions? Could OsNRT2.1 overexpression be used to improve rice yields under low nitrate conditions
Response 12: Thank you very much for your suggestion and we have added the discussion about the physiological relevance of our findings as “We first report that overexpression of OsNRT2.1 can show great root growth phenotype under low NO3- conditions through modulation of auxin transport and root elongation may depend on auxin transport but not the increase of nitrate influx.” In conclusion, please see line 436 in the revised version with trackers.
And we added more comments on the discussion part “this study provides clear evidences for the super phenotype of overexpression lines from the perspective of root.” Please see the line 431 in the revised version with trackers.

Reviewer 2 Report
The authors have performed phenotypic analysis on rice comparing NO3- and NH4+ treatments and exhibited the effect of NO3- in root system architecture (RSA). In addition they have explored the link between the effects of NO3- in RSA, auxin transport and response and the nitrate transporter OsNRT2.1.
My comments are the following:
- Introduction should include also brief information about auxin transport (PINs) and it s effect in root growth. Also more information about nitrogen supply (NH4NO3, NH4+, NO3-) and plant adaptation mechanisms should be introduced.
- Figure 3 can be moved to Sup. Figures. Root phenotype is fully analyzed in the following Figure 4 and shoot phenotype is not the focus point of this manuscript.
- The presentation of the data is unclear because results for mock treatments are not included in the same Figure. For example Figure 2 (NH4NO3 is the mock for Figure 4 (NH4+ and NO3-) and Figure 4 (NO3- without NPA) is the mock for Figure 7 (NO3- with NPA).
- In summary and discussion it is mentioned that NO3- influx increased 23.8% in OE lines in NO3- with no NPA while the increase was 25.1% when NPA was added. The percentages mentioned do not match the graphs in Sup. Fig S2 ans S4. The increase of NO3- influx in OE compared to WT seems to be smaller under NPA treatment. Fig S2 and Fig S4 should be merged and transferred to main figures while a table with the raw data values should be added to the Sup Data.
Further question/comment in this is the following:
One of the conclusions (line 326-327) is that NO3- induces PIN expression as shown in Fig 5. How would you explain/discuss that while NPA treatment did not impact the increased NO3- influx in OE lines, - (was the difference significant? Compare and perform statistics between NO3- influx +/-NPA) – PIN expression was altered/reduced.
- In both Fig 1A (6th day) and 1C (6d) shoot seems to be significantly smaller in OE lines. In contrast Fig 1D (7d) root is shown to be significanlty longer in OE lines but that s not exhibited in Fig1A (7th day).
- Specify in materials and methods or in figure legends what "total" stands for in y'y relevant charts (eg Fig 2B, D, G, H) . Is it a sum or average?
- Some reference or explanation/specification on why using 0.5mM as concentration for NO3- and NH4+ treatments. The lack of response to NH4+ could be concentration-related and a dose response experiment would be required before concluding that it does not affect growth and polar auxin transport (PAT).
-In Figure 5B OsPIN1b expression in OE lines seems and is mentioned in the text to be significantly upregulated under NO3-. However this is not displayed in the chart.
- Figures 5 and 6 should be supplemented with PIN expression in OE lines under NH4NO3 (respective control experiment).
- There are two relevant publications that should be discussed and compared with the data presented here. Sun et al., 2018_ Nitric Oxide Affects Rice Root Growth by Regulating Auxin Transport Under Nitrate Supply and Song et al., 2013_ Auxin distribution is differentially affected by nitrate in roots of two rice cultivars differing in responsiveness to nitrogen. Sun et al., has demonstrated a link between polar auxin transport and seminal root elongation which is also explored in this work (Figure 6A and C). In the same paper they also compare DR5 expression under NH4+ and NO3- treatments. Data should be mentioned, discussed and compared. In Song et al., it is shown that DR5 signal was increased under NH4 treatment with NPA addition compared to NH4 without NPA. This is opposite to the result presented in Figure 6B. Result should be compared and discussed.
- Overall this work is indicating a link between NO3- treatment, OsNTR2, auxin transport, auxin response and RSA. Most of these links have been established before. For example, Zhang et al 1999 displayed a that NO3- effect on RSA requires auxin response and Sun et al., 2018 displayed that the NO3- effect on RSA demands active polar auxin transport (PAT).
Therefore the novelty of the current work is the involvement of OsNTR2.1 in this mechanism and that OsNRT2.1 promoting effect on root growth upon NO3- treatment requires active PAT.
It is needed to strengthen the conclusions on this aspect and support the link between OsNTR2.1-hence the effect of NO3- as a signal- regulating PIN expression/auxin transport and to do so more data are needed.
For example what is the NO3- influx, PIN expression, auxin responsive genes expression (and/or DR5 signal) and respective RSA in when OsNTR2.1 is knocked out/down. Such results could be either confirmed or alternatively shown by application of chemical treatment blocking/reducing NO3- influx in rice (relevant chemicals can be used and/or cytokinins since they have been shown to repress AtNTR2.1 expression). It will be also nice to see DR5 expression or auxin responsive genes (eg AXR4) in OE lines.
Author Response
Point 1: Introduction should include also brief information about auxin transport (PINs) and its effect in root growth. Also, more information about nitrogen supply (NH4NO3, NH4+, NO3-) and plant adaptation mechanisms should be introduced.
Response 1: Thank you for your suggestion. We have added these contents in Introduction. Please see the lines from 40 to 49 and from 75 to 84 in the revised version with trackers.
Point 2: Figure 3 can be moved to Sup. Figures. Root phenotype is fully analyzed in the following Figure 4 and shoot phenotype is not the focus point of this manuscript.
Response 2: Yes, your suggestion is fine. We have changed figure 3 to supplementary data in Figure S2.
Point 3: The presentation of the data is unclear because results for mock treatments are not included in the same Figure. For example, Figure 2 (NH4NO3 is the mock for Figure 4 (NH4+ and NO3-) and Figure 4 (NO3- without NPA) is the mock for Figure 7 (NO3- with NPA).
Response 3: Thank you for your suggestion. We have redesigned these figures according to your request. Please see the figures 2 and 3.
Point 4: In summary and discussion, it is mentioned that NO3- influx increased 23.8% in OE lines in NO3- with no NPA while the increase was 25.1% when NPA was added. The percentages mentioned do not match the graphs in Sup. Fig S2 and S4. The increase of NO3- influx in OE compared to WT seems to be smaller under NPA treatment. Fig S2 and Fig S4 should be merged and transferred to main figures while a table with the raw data values should be added to the Sup Data.
Response 4: Thank you for your suggestion. We added the all raw data values in Table S2 with two-way ANOVA analysis for all the data under different treatment of all lines.
Actually, we have published this data (nitrate influx without NPA) in Figure S2 of reference 47 last year, which we carried with these experiments (with NPA treatment) at same time. We are very sorry for not making this clear in last version. We think even the Fig S2 has been published but it will be benefiting the reader to understand the whole story in this manuscript, if we could put it in supplemental data. Now we made a citation [47] in the results to explain the source of this data, please see the lines from 325 to 330 and from 353 to 360 in the revised version.
And as your suggestion, we moved the original Fig S4 (with NPA) into main figure. Thank you very much.
Point 5: Further question/comment in this is the following:
One of the conclusions (line 326-327) is that NO3- induces PIN expression as shown in Fig 5. How would you explain/discuss that while NPA treatment did not impact the increased NO3- influx in OE lines, - (was the difference significant? Compare and perform statistics between NO3- influx +/-NPA) – PIN expression was altered/reduced.
Response 5: Your suggestion is very nice. However, when we pooled the all the NPA treatment on 15N influx in Table S2, as your suggestion above, we found NPA did reduce the nitrate influx in both WT and over expression lines. Just compared with WT, the super pattern of nitrate uptake is still existing in over expression lines, under NPA treatments. Therefore, the PIN gene expression was reduced by the reduction in nitrate influx in WT by NPA treatment. However, the decrease pattern of the expression of OsPIN was much huge than that in nitrate influx.
Point 6: In both Fig 1A (6th day) and 1C (6d) shoot seems to be significantly smaller in OE lines. In contrast Fig 1D (7d) root is shown to be significantly longer in OE lines but that’s not exhibited in Fig1A (7th day).
Response 6: From the figure 1A (7th day), we can observe that there are some adventitious roots in overexpression lines but not very much in WT. Therefore, we measured the root length including seminal roots and adventitious roots.
Point 7: Specify in materials and methods or in figure legends what "total" stands for in y'y relevant charts (eg Fig 2B, D, G, H). Is it a sum or average?
Response 7: We got these indicators by root scanners. The total root length represents the sum of seminal roots, adventitious roots and all lateral roots. The total adventitious roots are the sum of all adventitious root length in every plant. The total LR length in seminal roots are the sum of lateral roots length in seminal roots of in every plant. The total LR length in adventitious roots are the sum of lateral roots length in adventitious roots of every plant. Please see the renewed method lines from 149 to 152, Figure 3 legend and Figure 6 legend in the revised version with trackers.
Point 8: Some reference or explanation/specification on why using 0.5mM as concentration for NO3- and NH4+ treatments. The lack of response to NH4+ could be concentration-related and a dose response experiment would be required before concluding that it does not affect growth and polar auxin transport (PAT).
Response 8: First, the expression of OsNRT2.1 was increased to improve nitrate absorption function in OE rice lines by 0.5mM NO3- treatment. So, we set 0.5mM as treatment. Second, Sun et al., 2018 found the lateral roots number and seminal roots length of rice even under 5mM NO3- condition were better than rice under 5mM NH4+ treatment. And, the IAA concentration in root tips and lateral root zone under were also higher under 5mM NO3- condition than 5mM NH4+ treatment in this paper. Therefore, we think the lack of response of NH4+ is not due to dose fact.
Point 9: In Figure 5B OsPIN1b expression in OE lines seems and is mentioned in the text to be significantly upregulated under NO3-. However, this is not displayed in the chart.
Response 9: I’m sorry about this. We mislabeled it. And, we have revised the figure 4B.
Point 10: Figures 5 and 6 should be supplemented with PIN expression in OE lines under NH4NO3 (respective control experiment).
Response 10: As many papers [76,77,78] have reported that nitrate can affect the root growth and the emergence of lateral root, only compared with the ammonium as control, we designed this experiment based on these references.
Point 11: There are two relevant publications that should be discussed and compared with the data presented here. Sun et al., 2018_ Nitric Oxide Affects Rice Root Growth by Regulating Auxin Transport Under Nitrate Supply and Song et al., 2013_ Auxin distribution is differentially affected by nitrate in roots of two rice cultivars differing in responsiveness to nitrogen. Sun et al., has demonstrated a link between polar auxin transport and seminal root elongation which is also explored in this work (Figure 6A and C). In the same paper they also compare DR5 expression under NH4+ and NO3- treatments. Data should be mentioned, discussed and compared. In Song et al., it is shown that DR5 signal was increased under NH4 treatment with NPA addition compared to NH4 without NPA. This is opposite to the result presented in Figure 6B. Result should be compared and discussed.
Response 11: Your suggestion is very good. We added these contents in Discussion part. Please see the lines from 389 to 393 and from 407 to 410 in the revised version with trackers.
Point 12: Overall this work is indicating a link between NO3- treatment, OsNTR2, auxin transport, auxin response and RSA. Most of these links have been established before. For example, Zhang et al 1999 displayed a that NO3- effect on RSA requires auxin response and Sun et al., 2018 displayed that the NO3- effect on RSA demands active polar auxin transport (PAT). Therefore, the novelty of the current work is the involvement of OsNTR2.1 in this mechanism and that OsNRT2.1 promoting effect on root growth upon NO3- treatment requires active PAT. It is needed to strengthen the conclusions on this aspect and support the link between OsNTR2.1-hence the effect of NO3- as a signal- regulating PIN expression/auxin transport and to do so more data are needed. For example, what is the NO3- influx, PIN expression, auxin responsive genes expression (and/or DR5 signal) and respective RSA in when OsNTR2.1 is knocked out/down. Such results could be either confirmed or alternatively shown by application of chemical treatment blocking/reducing NO3- influx in rice (relevant chemicals can be used and/or cytokinin since they have been shown to repress AtNTR2.1 expression). It will be also nice to see DR5 expression or auxin responsive genes (eg AXR4) in OE lines.
Response 12: Thank you for your suggestion, we crossed DR5 lines with OE lines last year and got OE3 with DR5 seeds just recently. Now we added the OE3 with DR5 marker line data in Figure S5 to show the effect of NPA on DR5 expression in OE3 lines. The results showed NPA also significantly reduced DR5 signal. However, as the seeds are limited, therefore we have not carried the same experiments on gene expression.
And as your suggestion, now we pointed out the novelty of our finding in conclusion as “We first report that overexpression of OsNRT2.1 can show great root growth phenotype under low NO3- conditions through modulation of auxin transport and root elongation may depend on auxin transport but not the increase of nitrate influx.” Please see lines from 438 to 441 in the revised version with trackers.

Round 2
Reviewer 2 Report
The manuscript is very good and most of the comments I made have been took into consideration. The figures look beautifully clear now and the references that were missing are now added.
My remaining concern is the "point 12":
Point 12: Overall this work is indicating a link between NO3- treatment, OsNTR2, auxin transport, auxin response and RSA. Most of these links have been established before. For example, Zhang et al 1999 displayed a that NO3- effect on RSA requires auxin response and Sun et al., 2018 displayed that the NO3- effect on RSA demands active polar auxin transport (PAT). Therefore, the novelty of the current work is the involvement of OsNTR2.1 in this mechanism and that OsNRT2.1 promoting effect on root growth upon NO3- treatment requires active PAT. It is needed to strengthen the conclusions on this aspect and support the link between OsNTR2.1-hence the effect of NO3- as a signal- regulating PIN expression/auxin transport and to do so more data are needed. For example, what is the NO3- influx, PIN expression, auxin responsive genes expression (and/or DR5 signal) and respective RSA in when OsNTR2.1 is knocked out/down. Such results could be either confirmed or alternatively shown by application of chemical treatment blocking/reducing NO3- influx in rice (relevant chemicals can be used and/or cytokinin since they have been shown to repress AtNTR2.1 expression). It will be also nice to see DR5 expression or auxin responsive genes (eg AXR4) in OE lines.
For example:
Abstract: Line 31-34: “These results indicated that NO3 could be used as a signal substance to induce the expressions…”
I don t agree that with the data presented here you can draw the conclusion about NO3 being the signal substance that causes increase of PIN expression and respective increase in root elongation unless you perform some of the additional experiments mentioned above (point 12).
However there are several nice conclusions from the data presented here. But more care is needed with words and statements for what is a conclusion deriving from the data in this manuscript.
You can conclude that in OE lines PIN expression is reduced and the roots are longer. There is proof that NTR2.1 increase NO3 influx (which you have already published before 47) and that NPA here is significantly reducing this influx rate. There is proof that when OsNRT2.1 is overexpressed there are longer roots (total), altered RSA and reduced PIN expression. So these conclusions can be supported by the data shown in this work. You can put more focus on the nice detailed RSA results you got. For example, in Figure 3 it is observed that while OE lines have longer seminal roots under NO3 their respective LR number is not increased. That would indicate there are more LR in seminal roots following NO3 treatment which is in agreement with other plant species.
This is something that need to be taken care of throughout the text (discussion) not only in abstract.
One last comment:
In Table S2 you now show that NPA treatment affects significantly the NO3- influx in OE lines. Would that mean that NRT2.1-related import of NO3- is partially auxin dependent? The conclusion about the influx rates (23.8% and 25.1%) with and without NPA respectively remained the same in the abstract. Are these the correct percentages according to Table S2?
Author Response
Point 1: My remaining concern is the "point 12":
Point 12: Overall this work is indicating a link between NO3- treatment, OsNTR2, auxin transport, auxin response and RSA. Most of these links have been established before. For example, Zhang et al 1999 displayed a that NO3- effect on RSA requires auxin response and Sun et al., 2018 displayed that the NO3- effect on RSA demands active polar auxin transport (PAT). Therefore, the novelty of the current work is the involvement of OsNTR2.1 in this mechanism and that OsNRT2.1 promoting effect on root growth upon NO3- treatment requires active PAT. It is needed to strengthen the conclusions on this aspect and support the link between OsNTR2.1-hence the effect of NO3- as a signal- regulating PIN expression/auxin transport and to do so more data are needed. For example, what is the NO3- influx, PIN expression, auxin responsive genes expression (and/or DR5 signal) and respective RSA in when OsNTR2.1 is knocked out/down. Such results could be either confirmed or alternatively shown by application of chemical treatment blocking/reducing NO3- influx in rice (relevant chemicals can be used and/or cytokinin since they have been shown to repress AtNTR2.1 expression). It will be also nice to see DR5 expression or auxin responsive genes (eg AXR4) in OE lines.
For example:
Abstract: Line 31-34: “These results indicated that NO3 could be used as a signal substance to induce the expressions…”
I don’ t agrees that with the data presented here you can draw the conclusion about NO3 being the signal substance that causes increase of PIN expression and respective increase in root elongation unless you perform some of the additional experiments mentioned above (point 12).
However, there are several nice conclusions from the data presented here. But more care is needed with words and statements for what is a conclusion deriving from the data in this manuscript.
You can conclude that in OE lines PIN expression is reduced and the roots are longer. There is proof that NTR2.1 increase NO3 influx (which you have already published before 47) and that NPA here is significantly reducing this influx rate. There is proof that when OsNRT2.1 is overexpressed there are longer roots (total), altered RSA and reduced PIN expression. So, these conclusions can be supported by the data shown in this work. You can put more focus on the nice detailed RSA results you got. For example, in Figure 3 it is observed that while OE lines have longer seminal roots under NO3 their respective LR number is not increased. That would indicate there are more LR in seminal roots following NO3 treatment which is in agreement with other plant species.
This is something that need to be taken care of throughout the text (discussion) not only in abstract.
Response 1: Your suggestions are very good. Thank you very much! So, we revised the abstract and discussion. Please see the Line 31-34 and Line 352-355 with trackers.
Point 2: One last comment:
In Table S2 you now show that NPA treatment affects significantly the NO3- influx in OE lines. Would that mean that NRT2.1-related import of NO3- is partially auxin dependent? The conclusion about the influx rates (23.8% and 25.1%) with and without NPA respectively remained the same in the abstract. Are these the correct percentages according to Table S2?
Response 2: Yes, your point is right. In Table S2, WT and all rice overexpression lines decreased the 15NO3- influx rate with NPA treatment. So, these data suggested NRT2.1-related import of NO3- may be partially auxin dependent. For the percentages in Table S2, there are deviations in different calculation methods. We agreed with you and the percentages were recalculated by the average data statistics in Table S2. We have changed this percentage value in the text with track. Besides, we revised the text according to your suggestion. Please see the Line31-34, Line342-344 and Line352-355 with trackers. Thank you very much!
